# Exploring the association between sick child healthcare utilisation and health facility quality in Malawi: a cross-sectional study

Lingrui Liu,[1,2] Hannah H Leslie,[3] Martias Joshua,[4] Margaret E Kruk[3]

[1]Department of Health Policy and Management, Yale School of Public Health, New Haven, Connecticut, USA
[2]Global Health Leadership Initiative, Yale University, New Haven, Connecticut, USA
[3]Department of Global Health and Population, Harvard T.H. Chan School of Public Health, Boston, Massachusetts, USA
[4]Malawi Ministry of Health, Lilongwe, Malawi

**Correspondence to**
Dr Lingrui Liu;
lingrui.liu@yale.edu

## ABSTRACT

**Objective** Increasing the availability of basic healthcare services in low-and middle-income countries is not sufficient to meet the Sustainable Development Goal target for child survival in high-mortality settings, where healthcare utilisation is often inconsistent and quality of care can be poor. We assessed whether poor quality of sick child healthcare in Malawi is associated with low utilisation of sick child healthcare.

**Design** We measured two elements of quality of sick child healthcare: facility structural readiness and process of care using data from the 2013 Malawi Service Provision Assessment. Overall quality was defined as the average of these metrics. We extracted demographic data from the 2013–2014 Malawi Multiple Indicator Cluster Survey and linked households to nearby facilities using geocodes. We used logistic regression to examine the association of facility quality with utilisation of formal health services for children under 5 years of age suffering diarrhoea, fever or cough/acute respiratory illness, controlling for demographic and socioeconomic characteristics. We conducted sensitivity analyses (SAs), modifying the travel distance and population—facility matching criteria.

**Setting and population** 568 facilities were linked with 9701 children with recent illness symptoms in Malawi, of whom 69% had been brought to a health facility.

**Results** Overall, facilities showed gaps in structural quality (62% readiness) and major deficiencies in process quality (33%), for an overall quality score of 48%. Better facility quality was associated with higher odds of utilisation of sick child healthcare services (adjusted ORs (AOR): 1.66, 95% CI: 1.04 to 2.63), as was structural quality alone (AOR: 1.33, 95% CI: 0.95 to 1.87). SAs supported the main finding.

**Conclusion** Although Malawi's health facilities for curative child care are widely available, quality and utilisation of sick child healthcare services are in short supply. Improving facility quality may provide a way to encourage higher utilisation of healthcare, thereby decreasing preventable childhood morbidity and mortality.

## Strengths and limitations of this study

► Using the spatial geocodes, this study linked the health system (2013 Malawi Service Provision Assessment) and the household (2013 Malawi Multiple Indicator Cluster Survey (MICS)) datasets to investigate the role of quality in sick child healthcare utilisation in Malawi.

► This study relied on internationally endorsed guidelines to define quantifiable quality of care measures.

► Multiple sensitivity analyses confirmed the findings.

► However, matching strategy of linking a sick child to a health facility may not completely reflect actual behaviour.

► We acknowledged that the MICS survey data on symptoms of illness do not provide sufficient specificity on illness severity and thus were unable to determine that all children with these symptoms in fact require formal healthcare at health facilities.

## INTRODUCTION

The global health community has achieved notable gains in the Millennium Development Goals (MDGs) era. However, elimination of preventable and treatable child mortality remains an urgent global health priority in the coming decade.[1 2] Over the past decade, under the umbrella goal of universal health coverage (UHC),[3 4] health policies have focused on the expansion of coverage of essential health interventions and basic services in sub-Saharan Africa (SSA) and other low-income regions. Due to the rising recognition that, without improving quality of care in the health system, improved access to healthcare alone would not achieve expected health outcomes, the global health community has begun to focus on improvement of healthcare quality.[5 6] Large expansions of health facility networks have been attained in SSA and other low-income regions, and yet, utilisation of available resources for care of sick children under 5 years remains low, resulting in inadequate care-seeking for children with diarrhoea, malaria or pneumonia.[7–10] Malawi, a SSA country whose government has declared reduction of preventable children-under-five mortality as a national priority, achieved the MDG targets

for improved child health.[11] However, healthcare utilisation for conditions treatable by the health system is low, as data from the most recent Demographic and Health Survey (DHS) 2015 in Malawi show.[12] Although healthcare facility utilisation increased steadily from 2000 to 2015, the proportion of sick children under 5 years with symptoms of acute respiratory infection (ARI), fever and diarrhoea who were taken to a health provider for treatment within 48 hours of symptom onset, remained insufficient (51%, 46% and 66%, respectively).[7] These utilisation patterns occur despite wide availability of child health services (including outpatient curative care, child growth monitoring and child vaccination) in Malawi's health facilities,[12] with a median distance to the nearest health facility of 5 km and an estimated median travelling time of 75 min.[13] There is a growing recognition that, besides geographical access, poor quality of care could be a supply-side barrier to UHC, deterring patients from obtaining treatment and influencing family decisions to use or avoid services, which would ultimately impact health outcome gains.[14–17]

Although Malawi has achieved substantial improvement in coverage (utilisation by those in need) for curative care in children, as estimated from household survey data, quality remains weak.[18 19] Moreover, a recent multi-country study demonstrated that the duration and the content of sick child healthcare service were inadequate.[20] Another recent study echoed that care quality in pneumonia diagnosis for a sick child in Malawi is poor, with low guideline adherence to the Integrated Management of Childhood Illness (IMCI) criteria.[21] Most existing evaluation studies on the quality of sick child healthcare delivered at health facilities in Malawi have aimed to describe the state of quality but few have tackled the question of whether poor quality dissuades families from using the health system when a child falls ill. In the area of reproductive health, one study suggested a positive association between the perceived quality of reproductive care at the facility last attended and the family utilisation of immunisation and treatment services for their children at these health facilities, as observed within facilities in Kisumu Municipality in Western Kenya.[22] Another recent research from a health intervention programme in Ntcheu district in Malawi identified that quality of perinatal care provided is an important predictor of both women's use and satisfaction with such services received.[23] Furthermore, earlier literature on bypassing for facility delivery in rural Africa has documented that quality of care is influencing clinic choice.[24] Although these studies suggested a positive association between respondent perception of quality and utilisation, it would be worthwhile to investigate this relationship at a broader population level (eg, using national household-representative data), as well as through use of the most recent health facility assessment standardised surveys, to explore the relationship.

Policymakers need evidence on what health system attributes (eg, quality of sick child care) may influence primary care utilisation. Malawi is focusing on health sector strategies to improve child health and well-being, and has embarked on its second Health Sector Strategic Plan (HSSP 2011–2016), making this an opportune time to seek evidence on the health system factors (eg, quality) that best promote utilisation of sick child healthcare services. In this paper, we examine the association between quality of care in health facilities in Malawi and utilisation of sick child healthcare by caregivers. This study is one of the few which we are aware that focuses on the contribution of facility quality to utilisation of child healthcare services in high-disease-burden settings.[25]

## METHODS

### Study sample

Malawi is one of the smallest and most densely populated countries in SSA, made up of a predominantly young population, of which almost half is aged 15 years or younger, and more than 20% children under 5 years. With a total population of 17 million and a gross domestic product per capita of US$1200 in 2015, approximately 52% of Malawi's population lives below the international poverty line ($1.90 per person per day in 2011 purchasing power parity dollars), with more than 80% inhabiting rural regions.[26] Health facilities that provide child health services in the formal sector of Malawi include hospitals, health centres, clinics, dispensaries and health posts.

The analysis in this research combined data from multiple sources. To obtain information on the facility quality of child health services, we used the Malawi 2013–2014 Service Provision Assessment (SPA), a comprehensive census of all formal-sector health facilities conducted by the DHSs Programme.[12] SPA is a standardised cross-sectional survey of public and private health facilities, which includes a facility-level audit, clinical observation protocols, patient exit interviews and healthcare provider interviews. The health facility audit was conducted in each health facility visited using a standardised inventory questionnaire. At least one health worker was interviewed in a sampled health facility. Interviewers also observed a sample of patient–provider clinical care service interaction processes, as well as interviewing patients observed receiving care on their exit from the health facility. The SPA survey does not report or examine clinical outcomes. In this study, we examined the quality of outpatient sick child curative care delivered at health facilities in Malawi, excluding child vaccination services and child growth monitoring services. We limited our sample of facilities to those facilities (including hospitals, health centres, health posts, dispensaries) which provided outpatient sick child curative care services.

To obtain household information and care-seeking for children under 5 years of age, we used the 2013–2014 Malawi Multiple Indicator Cluster.[27] Malawi 2013 Multiple Indicator Cluster Survey (MICS) is a household survey conducted to assess health outcomes among a nationally representative sample of the population, employing a multi-stage sampling strategy. Enumeration areas (EAs)

were sampled within the strata of district and urban versus rural location, and then households were identified within EAs. With a systematic sample of 25 households drawn in each sample cluster, a total sample of 1140 EAs and 28 479 households were selected for the Malawi 2013 MICS. We limited our sample of households to those in which caregivers reported children under 5 years of age who had an episode of diarrhoea, symptoms of ARI or fever during the 2 weeks prior to the survey. Although these three specific medical conditions cannot cover the full range of illnesses of children under 5 years of age, malaria, diarrhoea and pneumonia remain leading causes of death among children under 5 years of age and are the most consistent indicators for child disease burden across the globe, with the vast majority occurring in low- and middle-income countries (LMICs).[28] These conditions combined are relevant in epidemiological predictions and represent demand for sick child healthcare services in Malawi.

## Measure of quality of child health services delivered at health facilities in Malawi

To date, there has not been a single uniform set of measures on quality definition and metrics.[15 29] The Institute of Medicine report *Crossing the Quality Chasm* identified six dimensions to measure quality: safe, effective, patient-centred, timely, efficient and equitable.[30] We consolidated these dimensions with Donabedian's[31] quality of care framework of structure, process and outcome. The structure elements describe the characteristics of the environment in which healthcare is provided, which exist before the care takes place. The structural inputs of a health system indicate the capability and serviceability of a health system under which care occurs.[32] Process features include two key components: technical interventions (appropriate delivery of clinical procedures following clinical guidelines, eg, WHO IMCI guidelines in child curative care services) and interpersonal interactions between users and a healthcare system. Outcomes refer to the final consequences of healthcare, such as the children-under-five mortality rate.[32] In this study, available data did not specify patient outcomes for the children experiencing recent illness.

Using Donabedian's framework, we constructed an index of structural quality based on the facility audit and an index of process quality based on the observation of healthcare delivered. We reviewed the WHO Service Availability and Readiness Assessment (SARA) to identify relevant structural quality indicators and the WHO IMCI for expected clinical actions and matched these to items available in the SPA survey. A total of 29 items on three domains were identified matching the SARA health facility readiness items (general readiness and readiness for the specific child curative care service) to assess the structure quality, using data extracted from the SPA health facility audits: (1) infrastructure (eg, water, electricity, ambulance, etc); (2) equipment, essential supplies and medications (eg, gloves, sharps, medications storage,

daily computer updates, etc); (3) staffing and management (eg, supervision provided, staff training, etc). Moreover, 18 indicators were identified matching the IMCI items to assess the observed clinical care process quality. This process quality index covers assessment of clinical history, routine examination and counselling on danger signs. We first averaged structural indicators within facility. We then averaged process indicators within observation. Furthermore, we averaged these clinical observations for sick child visits within each facility to obtain the facility-level process quality score, using a rescaled weight for each clinical observation to reflect the sampling probability of patients within facilities. We further averaged the facility-level structure and process quality score to arrive at a facility-level overall quality score to measure average performance on all indicators. For each quality index, indicators were averaged to provide a facility summary score from 0 to 1. Multiple imputation was applied to address the missingness of individual indicators for facilities without sick child observations (up to 19%; see online supplementary appendix table 1 for details), using the R Amelia package.[33] Considering the multiple imputation assumption that missingness is random conditional on the covariates, we included four covariates in our imputation: facility type (eg, central hospital, district hospital, health centres, etc), facility managing authority (eg, government, private, non-governmental organisation, mission/faith-based, etc), districts and urban/rural.

### Sick child care utilisation

Utilising sick child healthcare was defined as a binary indicator that children with diarrhoea, fever or symptoms of ARI sought curative care at formal health facilities. Following the MICS report in identifying which response options were considered as formal care and being consistent with the facility types surveyed in SPA, we consider health facilities including hospitals, health centres, clinics, health posts and dispensaries, and did not include other sources of informal care (eg, traditional healers or shops).

### Covariates

We obtained data from the MICS on household socioeconomic status (household wealth index, urban/rural residence), and demographics of caretaker and ill child (child's age and gender, mother's education level) as well as child illness type. The household wealth index was calculated following standard procedures for the DHS and classified into quintiles by the MICS.

### Patient and public involvement

Patients and the public were not involved in the research design or planning of this study.

### Statistical analysis

We obtained the spatial location of all enumeration areas (EAs) for the MICS from the 2008 Malawi census data. Each sick child was linked to the geographical centroid of the associated household's EA using EA codes provided

by the Malawi National Statistical Office. SPA data include exact location of each facility. There was limited empirical evidence from Malawi regarding where children were taken to seek treatment for their illness. In this study, rather than assessing bypassing, we focused on whether having a high-quality facility nearby promotes household utilisation of sick child health services. We matched each sick child and his/her household to the single nearest facility, based on travel distance by road to the EA centroid of the sick child's household. To calculate the geographical distance, we executed four steps: first identified the closet eight facilities providing sick child healthcare using the geocoordinates for facilities (exact location) and household clusters (centroid); second, the road distance was calculated between cluster centroid and each facility based on the Google Maps Application Programming Interfaces (API), including linear distance to nearest road where coordinates are not directly on a road (road type is not incorporated in this calculation); third, we replaced road distance with linear distance if road distance cannot be calculated (eg, if there is no road on an island) or if road distance is less than linear distance; lastly, we identified the facility with minimum distance to the cluster. Steps 1, 3 and 4 were executed in Stata, and Step 2 was run in Python V.3.6.1. Based on prior studies suggesting that household distance to nearest health facility in Malawi is rarely greater than 50 km,[34–36] we excluded children whose nearest facility was over 50 km away.

Descriptive analyses of facility quality for sick child healthcare were first performed. We used logistic regression analyses of utilisation of sick child healthcare on the quality index and then adjusted for the covariates of interest. We used clustered standard errors to account for the non-independence of observations within EAs. To understand which element of the overall quality served best as a predictor for household utilisation of sick child healthcare, we separated overall facility quality into structural and process quality.

We further conducted a series of sensitivity analyses (SAs) to understand the robustness of our results. To understand whether the best performing facility may be more influential than the nearest facility, as long as it is still relatively accessible, we matched the household with the best performing facility within the buffer zone of a 5 km radius (direct distance from the household's EA centroid). Additionally, to test the sensitivity of catchment area definitions, we continued using the best performing facility to match with the households, but modified the original 5 km buffer zones to 10 km and 20 km direct distance radius from the household's EA centroid.

Statistical analyses were run in Stata (V.14.1), mapping was done using QGIS V.2.18 (Free Software Foundation, Boston, MA, USA) and geographical distances (eg, road travelling distances) were calculated based on Google Maps using Python V.3.6.1.

## RESULTS

### Participants

In the 2013 SPA, 977 of a total of 1066 surveyed health facilities (response rate: 92%) completed the assessment. Among these 977 facilities, 920 facilities (94%) offered sick child health services. Among these 920 facilities, 746 facilities had observations of sick child healthcare; process quality indicators were imputed for the 174 facilities that offer sick child services but did not have any observations of care. Completeness of each indicator is shown in online supplementary appendix table 1. There was no missingness for infrastructure, but minimal for equipment, essential supplies and medications, substantial for supervision (up to 14%), and moderate for process quality (up to 19%).

The 2013 MICS dataset included 18 981 children under 5 years of age with completed caretaker interviews (response rate: 98%). Among these responding households, 52% of children (n=9811) were reported by their caretakers as having symptoms of diarrhoea (n=4419, 45%), ARI (n=1438, 15%) or fever (n=7118, 73%). Among the 9811 children who had illnesses, 6679 children (68%) sought care at facilities (hospitals, health centres/clinics or health posts/dispensaries).

In our main matching strategy, 110 of the 9811 sick children were dropped due to the EAs for which locations are not available. Then, each of the 9701 sick children was matched to the child's single nearest facility of the 920 health facilities providing sick child care based on road travelling distance to the EA centroid of the sick child's household. In this step, 352 health facilities were dropped. Therefore, our main matching strategy yielded an analytical sample of 568 health facilities providing sick child healthcare services and 9701 children who were reported by their caregivers as having illness of diarrhoea, fever or ARI. A total of 8363 (86%) sick child consultations were actually observed in these selected facilities.

### Descriptive data

Table 1 details the characteristics of health facilities in our main analytic sample. Among the health facilities that were included in the main analytic sample, health centres (57%) and clinics (26%) were the most common. The bulk of health facilities had clinical officers as the highest level of provider present (79%). The number of facilities located in rural areas was about fourfold those in urban settings while about half were managed by government authority. Figure 1 shows the geographical distribution of all 2013 SPA health facilities and the ones included in our analytic sample, the 2013 MICS EAs, as well as the population density in Malawi.

Figures 2 and 3 detail the facility performance on a structural and process quality index, respectively. The average structural quality score for health facilities in Malawi providing sick child curative care services was 0.62 (SD: 0.14, range: 0.20–0.97) and the average process quality score was 0.33 (SD: 0.14, range: 0.04–0.78). The average overall quality score was 0.48 (SD: 0.10, range:

**Table 1** Facility characteristics and quality performance on sick child healthcare services in the analytic sample (Main Model: n=568)*

| | All facilities (568) | |
| --- | --- | --- |
| | n or mean | % or SD |
| Rural† | 445 | 78.4 |
| Urban | 123 | 21.6 |
| Public‡ | 300 | 53.0 |
| Private | 268 | 47.0 |
| Facility type | | |
| Hospital | 75 | 13.2 |
| Health centre | 323 | 56.9 |
| Clinic | 145 | 25.5 |
| Health post | 3 | 0.5 |
| Other (dispensaries) | 22 | 3.9 |
| Highest clinician on site | | |
| Medical doctor | 60 | 10.6 |
| Registered nurse | 8 | 1.4 |
| Enrolled nurse | 38 | 6.7 |
| Assistant medical officer | 9 | 1.6 |
| Clinical officer | 448 | 78.9 |
| Other health professional | 5 | 0.9 |
| Overall Quality Performance (mean, SD) | | |
| Structural quality | 0.62 | 0.14 |
| Process quality | 0.33 | 0.14 |
| Overall quality | 0.48 | 0.10 |

*In the main model (n=568), the analysis was restricted to facilities offering sick child healthcare that matched (within 50 km) to a household sampled in the Multiple Indicator Cluster Survey with a recently ill child.
†Facility is in rural area.
‡Facility is managed by government authority.

0.19–0.90). Facilities were commonly equipped with basic infrastructure, such as client waiting rooms and general facility cleanliness, while they still lacked light sources, electricity, toilet, ambulance and computer or internet access. Health facilities achieved high-level performance on essential supplies and medication readiness, while about 40% had sick child health service room infection control supplies. Within the staffing and management domain, nearly all facilities reported employing some form of supervision while a modest number performed well in routine quality assurance or else received inadequate staff training on IMCI child health services. Reporting client opinions was extremely rare. With the exception of high percentages in process quality indicators achieved for fever, coughing and temperature examination, all other indicators performed relatively modestly. The poorest indicator was providing counselling and examination for danger signs, oedema, maternal–infant transmission of HIV and ear pain.

Table 2 provides the characteristics of sick children in the analytic sample, of whom, 6679 children (69%) sought care at facilities during their illness while 3022 (31%) did not seek care. The average age of sick children was 29 months (SD: 16 months), with 16% younger than 1-year old. Half (50%) of the sick children were girls. The bulk of sick children had a perceived symptom of fever (64%), with ARI the least common (15%). The average road travelling distance to the nearest facility was about 5.8 km (SD: 4.7 km; median: 4.8 km).

### Main findings

Table 3 shows the results of the multivariable logistic regression models of the association between quality of care and sick child healthcare service utilisation in Malawi health facilities. Models 1 and 4 represent the base and fully specified models, respectively. Models 2 and 3 represent models adjusting for individual factors. Model 5 represents the fully specified model with the quality index separated into structural inputs and process quality.

The overall quality index of structural and process quality was a significant predicator of utilising formal health facilities for sick child healthcare services in Malawi. In the fully specified model (Model 4), the odds of utilising formal health facilities increase with increasing quality (adjusted OR (AOR): 1.66, 95% CI: 1.04 to 2.63). The magnitude of the effect increases slightly when adjusting for the control variables and remains significant in all models (Models 1–4). When separating the overall quality index into structural inputs and the process quality in the full specified model (Model 5), structural quality was a significant predictor (AOR: 1.33, 95% CI: 0.95 to 1.87) while process quality was positively but not significantly associated with utilisation (AOR: 1.25, 95% CI: 0.91 to 1.72). The overall quality remained as a significant predictor for utilising health facilities for sick child healthcare in the three SAs shown in online supplementary appendix table 2. These SA findings supported the association between facility quality and sick child healthcare utilisation even when using a large catchment area.

A child's type of illness and mother's education were found to be significant individual-level predictors of sick child healthcare utilisation. For children's illness type, in the fully specified model (Model 4) the results suggest higher odds of utilising health facilities for children with a reported fever (AOR: 1.17, 95% CI: 1.03 to 1.34) and symptoms of ARI (AOR: 1.78, 95% CI: 1.53 to 2.06), compared with those who were perceived to have diarrhoea. This indicates that the type of childhood illness (ie, diarrhoea, fever or ARI) is associated with motivating caregivers to utilise health facilities. For mother's education, the results suggest a gradient with increasing odds of utilising health facilities as the level of the mother's education increases. Compared with mothers who did not have primary education, mothers who had primary education and those had secondary or higher education had 30% (AOR: 1.30, 95% CI: 1.13 to 1.49) and 65% (AOR: 1.65, 95% CI: 1.38 to 1.97) higher odds of utilising

 

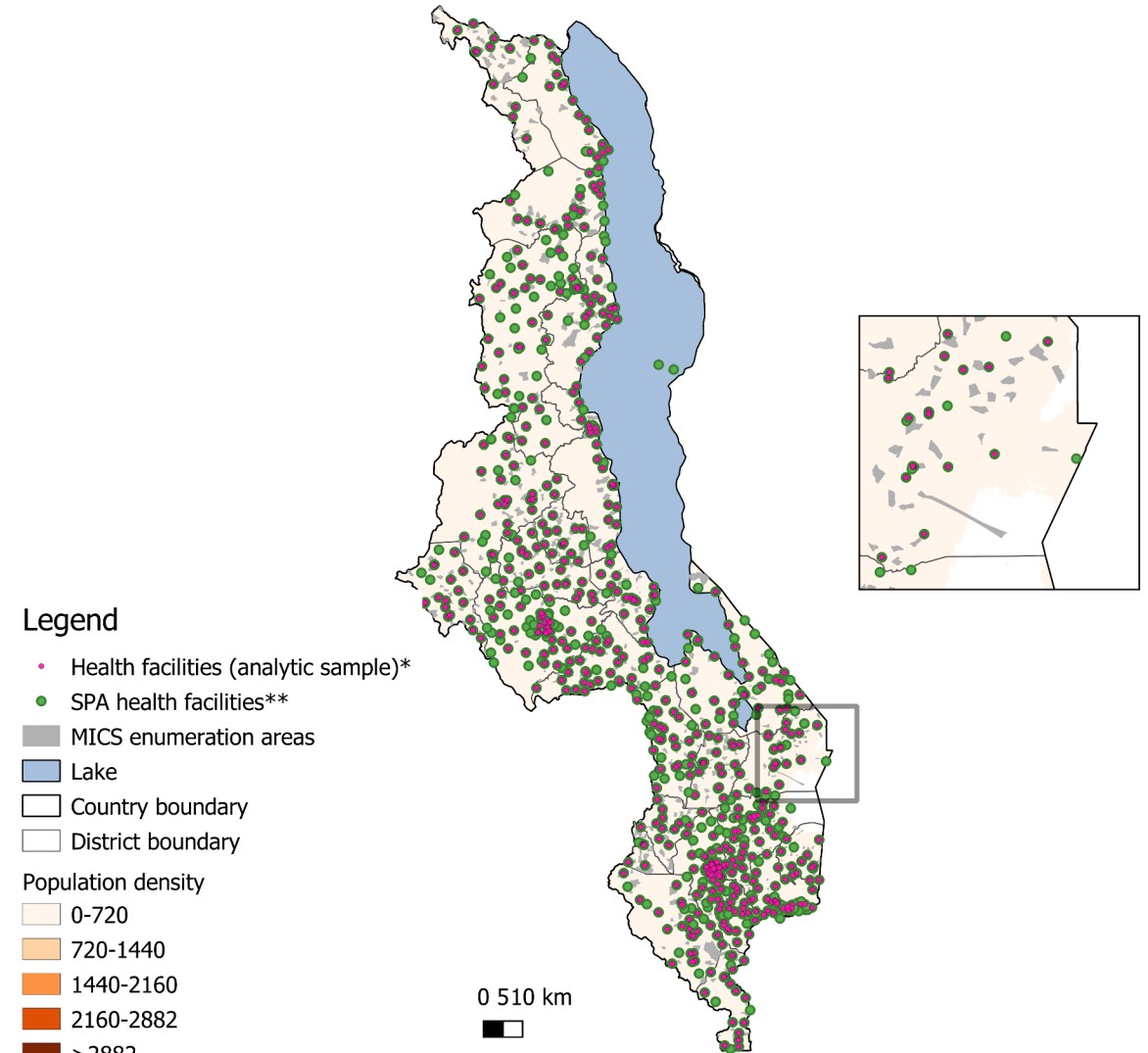

**Figure 1** Distribution of health facilities in Malawi. *Analytic sample (n=568). In the main model, the analysis was restricted to facilities offering sick child healthcare that matched (within 50 km) to a household sampled in the MICS with a recently ill child. **Malawi SPA health facilities (n=977): the health facilities completed the assessment in Malawi 2013 SPA. Data source: the map was created using the following data sources: (1) Population density: Worldpop.org (open access) https://www. worldpop.org/geodata/summary?id=123, WorldPop. 2017. Malawi 100m Population, Version 2. University of Southampton. DOI: 10.5258/SOTON/WP00538. (2) MICS enumeration areas: Malawi National Statistical Office. (3) Malawi shapefile and health facility location: The DHS Program.

health facilities, respectively, after controlling for other factors.

## DISCUSSION

In this paper, we sought to determine the influence of facility-based quality of healthcare on care-seeking behaviours of Malawian parents faced with childhood illness. We found that facility quality is an important predictor of parental decision-making regarding care-seeking for illnesses such as diarrhoea, fever and ARI among children under 5 years, using national data from Malawi. Our SAs varying the catchment areas consistently supported this association.

We found that about 69% of surveyed caregivers for sick children in our analytic sample utilised facility-based

healthcare services in Malawi, reflecting a similar prevalence of care-seeking reported by the most recent two Malawi DHS surveys.[7 37] Although Malawi has been one of the top three countries with highest prevalence of care-seeking for children with reported fever, diarrhoea or symptoms of ARI in African maternal and child health (MCH) priority countries, care-seeking for sick children under 5 years was still not optimal given the target of ending preventable deaths of children under 5 years by 2030 in Malawi HSSP.

Our findings confirmed past research showing major gaps in service readiness and provider competence in sick child healthcare. Like this work, others have noted poor provider performance, with fewer than half of clinical actions completed (eg, taking patient history,

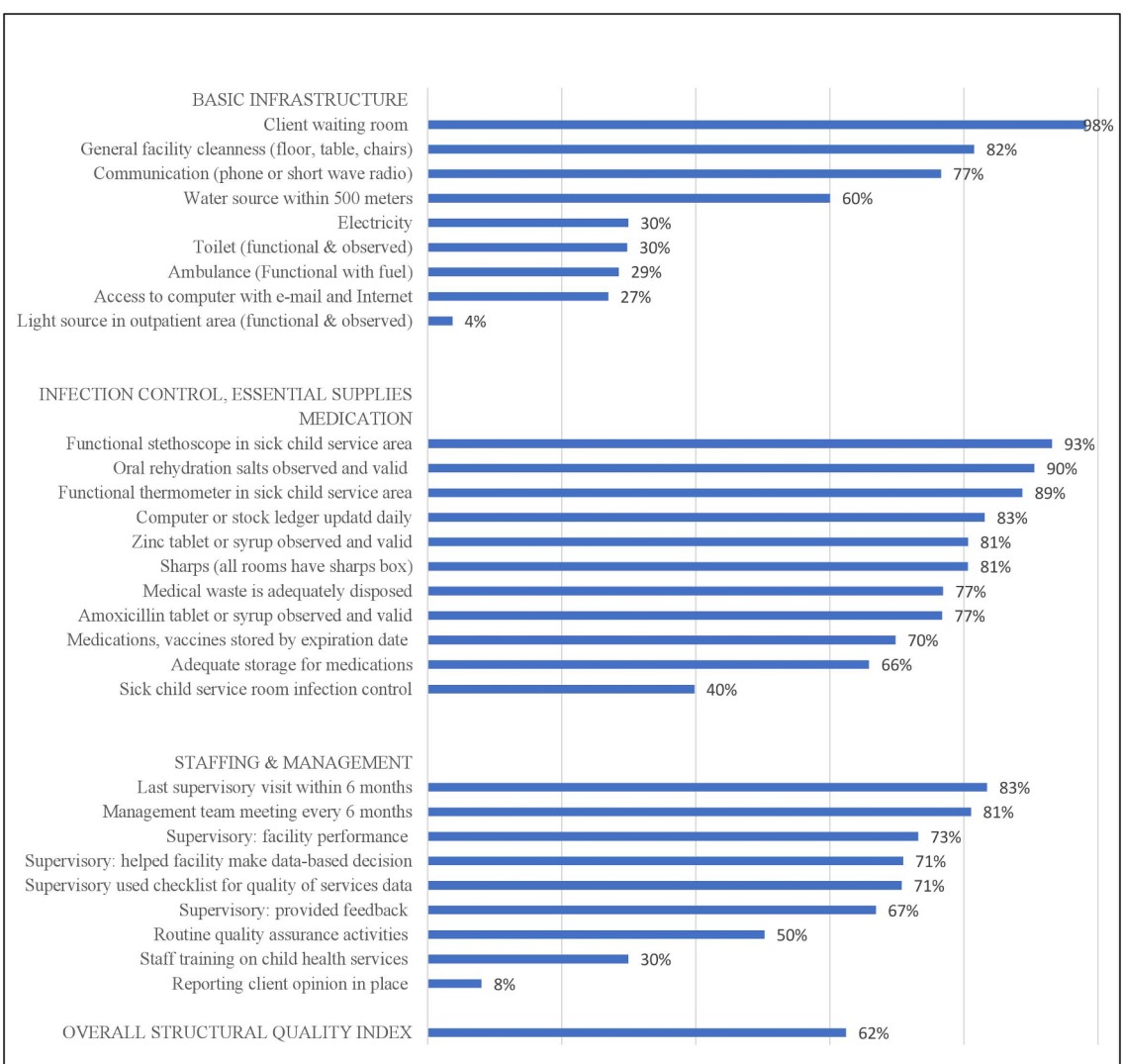

**Figure 2** Performance on facility structural quality index: percentage of facilities with key resources and services (n=568*). *In the main model (n=568), the analysis was restricted to facilities offering sick child healthcare that matched (within 50km) to a household sampled in the MICS with a recently ill child.

examination and counselling).[18 20] Moreover, guideline adherence was low, with a high rate of missed diagnosis among children (four out of every five children) with pneumonia.[38]

Our work advances on prior research in several ways. Past studies have largely focused on other factors in determining a caregiver's appropriate care-seeking for childhood illness, such as socio-demographic and household characteristics, social cultural factors, geographical access, travel time, health facility fees, insurance, health worker densities, etc.[39–43] Most prior studies on quality care for sick child healthcare services were specific evaluations of quality improvement interventions, such as IMCI community health worker programmes.[44–46] Few studies have examined the relationship of quality and healthcare utilisation on a national scale with regard to childhood illness in LMICs, with only one publication from Kenya attempting to construct the relationship between patient-perceived quality and the low attendance at the MCH services using local district data.

When disaggregating our quality measure into structural and process components, we found that structural readiness (facility infrastructure, equipment and staff) was a significant predictor of utilisation, whereas process quality (adherence to medical guidelines) was positively but not significantly predictive of utilisation. This finding suggests that individuals (caregivers) may weigh visible health facility characteristics (such as presence of client waiting room, general facility cleanness, equipment, drugs, etc) more than they weigh the actual clinical care service experience. Because of the asymmetry of information between providers and patients in the healthcare market, patients do not have full knowledge about what constitutes good quality; in this case, they may not be fully aware of the recommended components of clinical assessment for their child.[47]

This study had several strengths. First, the availability of exact spatial location data of the SPA health facilities and all EAs for the MICS from the 2008 Malawi census data provided a unique opportunity to examine health

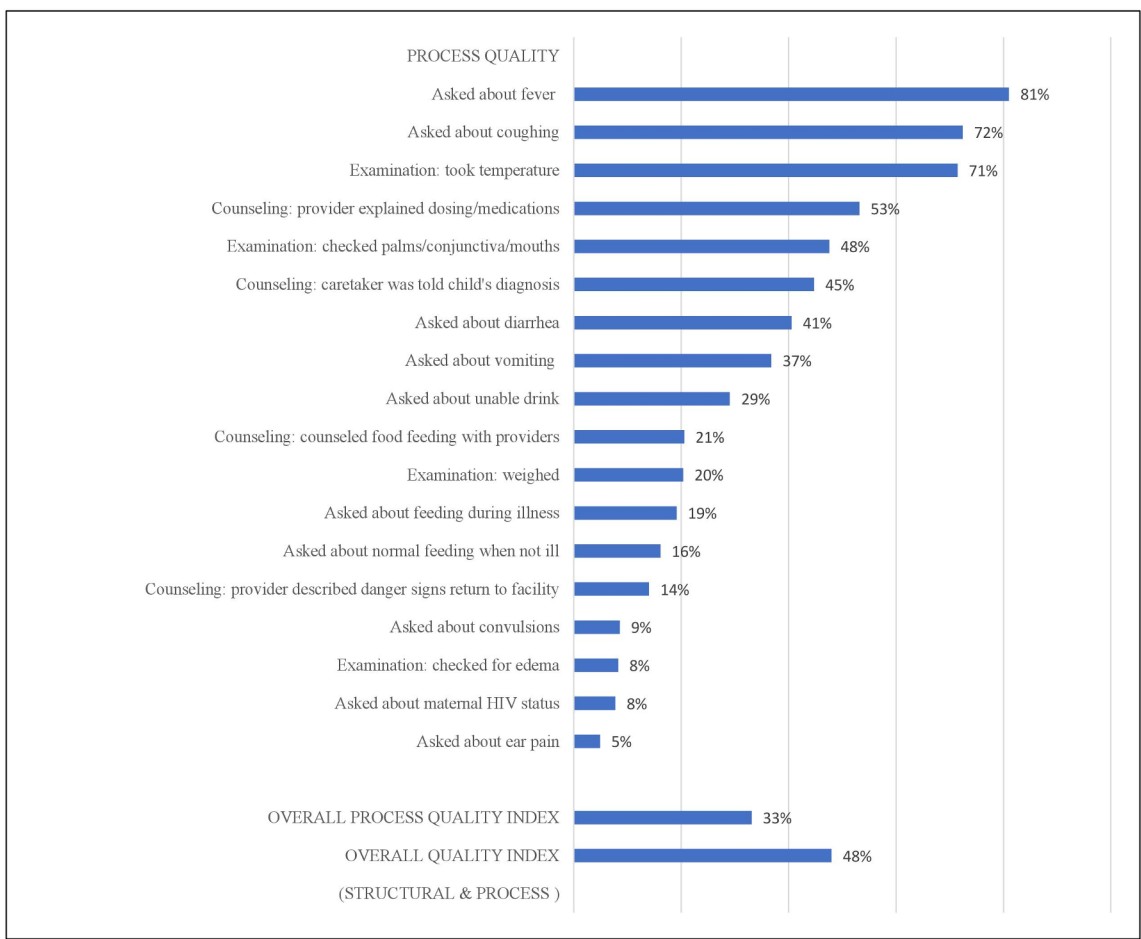

**Figure 3** Performance on facility process quality index: percentage of facilities with key resources and services (n=568*). *In the main model (n=568), the analysis was restricted to facilities offering sick child healthcare that matched (within 50km) to a household sampled in the MICS with a recently ill child.

system and population data together in concert to answer a policy-relevant question. By contrast, DHS household locations are displaced up to 5 km in rural areas, which precludes accurate matching to nearest facility. Second, we relied on WHO SARA and IMCI guidelines to define quality measures. Furthermore, multiple SAs confirmed that the findings were consistent across different facility matching specifications.

However, this study has several limitations. First, this study is based on cross-sectional data and is therefore unable to address causal claims. Second, given the available Malawi healthcare utilisation data (2013 MICS), we chose the most clinically relevant questions (variables) to indicate the demand by under-five children with sickness of diarrhoea, malaria or pneumonia. Our study focused on understanding the influence of quality on caregivers' decisions to utilise health facilities for their sick child. However, no patient outcome data were available, which prevented investigation of the linkage between facility quality and patient outcomes. Third, due to the data limitations, we extracted the quality index items available from the Malawi SPA dataset with reliance on the WHO SARA and IMCI, and therefore this analysis focused on specific health facility characteristics and interviewer-observed

clinical quality service, which however do not assess individual perceptions of healthcare quality, nor address the patient–provider interpersonal quality of care. Additionally, we acknowledged the common challenge of classifying childhood illness based on survey data in LMICs,[48 49] thereby our research was unable to capture the sufficient specificity on illness severity and identify all children with these symptoms who actually require formal healthcare at health facilities. Furthermore, in our multiple imputation strategy, we assumed that missingness of the health facility which provided sick child healthcare but had zero observations within a stratum like rural dispensaries is random. Our assumption was not that the missingness of these observations is randomly distributed across hospitals and dispensaries. Therefore, we included four covariates (facility type, facility managing authority, districts and rural/urban) in the multiple imputation. Moreover, in the real world, caretaker decisions in seeking care for a sick child, as well as where to seek healthcare, can be influenced by their perception of the quality of child health services at facilities recently visited. Our study used the facility-based patient data that captures the nearest facility to the patient's household and we were not able to identify the previous facility which the patient visited. Given

**Table 2** Characteristics of sick child in the analytic sample (main model: n=12 258)

| Variable | Total sick children (n=9701) | | Sought for care at facilities during illness (n=6679) | No care-seeking during illness (n=3022) |
| --- | --- | --- | --- | --- |
| | n | %* | n (%†) | n (%†) |
| Child age (in months) | | | | |
| ≤2 months | 162 | 1.7 | 88 (54.3) | 74 (45.7) |
| 2 months to 1 year | 1537 | 15.8 | 1052 (68.4) | 485 (31.6) |
| 1–5 years | 8002 | 82.5 | 5539 (69.2) | 2463 (30.8) |
| Child sex | | | | |
| Male | 4870 | 50.2 | 3420 (70.2) | 1450 (29.8) |
| Female | 4831 | 49.8 | 3259 (67.5) | 1572 (32.5) |
| Child sickness type | | | | |
| Diarrhoea | 2097 | 21.6 | 1284 (61.2) | 813 (38.8) |
| Fever | 6185 | 63.8 | 4352 (70.4) | 1833 (29.6) |
| Acute respiratory infection | 1419 | 14.6 | 1043 (73.5) | 376 (26.5) |
| Mother's education | | | | |
| None | 1287 | 13.2 | 817 (63.5) | 470 (36.5) |
| Primary | 7016 | 72.3 | 4828 (68.8) | 2188 (31.2) |
| Secondary or above | 1392 | 14.3 | 1031 (74.1) | 361 (25.9) |
| Household wealth quintile | | | | |
| Q1 (poorest) | 2380 | 24.5 | 1605 (67.4) | 775 (32.6) |
| Q2 | 2227 | 23 | 1515 (68.0) | 712 (32.0) |
| Q3 | 2140 | 22.1 | 1472 (68.8) | 668 (31.2) |
| Q4 | 1703 | 17.6 | 1205 (70.8) | 498 (29.2) |
| Q5 (richest) | 1251 | 13 | 882 (70.5) | 369 (29.5) |
| Road travelling distance (km) to nearest facility mean (SD, median) | 5.77 (4.7, 4.8) | | 5.66 (4.7, 4.7) | 6.00 (4.7, 4.9) |

*Column percentages.
†Row percentages.
‡n=9701, number of children who were perceived as sick in last 2 weeks by their caretakers that plausibly matched with their households' single nearest sick child health service facility in Service Provision Assessment . In the main model (n=568), the analysis was restricted to facilities offering sick child healthcare that matched (within 50 km) to a household sampled in the Multiple Indicator Cluster Survey with a recently ill child.

literature from other nations with comparable settings, in their first action, caregivers would seek healthcare at the nearest facility on recognising the child's symptom; however, they would choose to bypass their nearest facility when it lacked diagnostic equipment, drugs and skilled health workers, or had poor services.[22 50–52] In this study, we were not able to examine the influence of quality of the previous facility in relation to the caregiver's next health services facility choice, although these types of associations have been beneficial in prior studies in developed countries.[53] Thus, the matching strategy of linking a sick child to a health facility may not completely reflect actual behaviour.

### Policy implications

Our findings have several implications for policy and future research. Our study provides strong and direct empirical evidence that better quality health facilities are associated with increased healthcare utilisation for childhood illness, which, if care is sufficiently competent, can improve odds of survival from treatable conditions. As our findings suggested, the government of Malawi may consider the improvement of the health facility structure, including equipment, essential supplies, drug storage and availability, room cleanness and infection control. Visible improvements are most likely to attract caregivers in utilisation of the health facilities when they perceive childhood illness. However, beyond driving utilisation, provider clinical competence needs to improve as well if visits are to be converted into better health. Given the low level of provider performance we and others have documented, this will require structure system reforms, potentially including updating pre-service education and re-organising where healthcare is delivered for maximum gains.[6]

**Table 3** Regression results for the association between sick child healthcare utilisation and the overall quality (structural and process quality) of health service facility in Malawi†

| Main models | (1) | | (2) | | (3) | | (4) | | (5) | |
|---|---|---|---|---|---|---|---|---|---|---|
| Variables | ORs (p value) | 95% CI* | Adjusted ORs (p value) | 95% CI | Adjusted ORs (p value) | 95% CI | Adjusted ORs (p value) | 95% CI | Adjusted ORs (p value) | 95% CI |
| Overall quality | 1.53 (0.08) | (0.96 to 2.43) | 1.61 (0.05) | (1.01 to 2.56) | 1.67 (0.03) | (1.05 to 2.65) | 1.66 (0.03) | (1.04 to 2.63) | - | - |
| Structural quality | | | - | | - | | - | | 1.33 (0.10) | (0.95 to 1.87) |
| Process quality | | | - | | - | | - | | 1.25 (0.17) | (0.91 to 1.72) |
| Child age (Ref:≤2months) | | | | | | | | | | |
| 2 months to 1year | | | 0.99 (0.90) | (0.88 to 1.12) | 1.00 (1.00) | (0.89 to 1.13) | 1.00 (1.00) | (0.89 to 1.13) | 1.00 (1.00) | (0.89 to 1.13) |
| 1–5years | | | 1.98 (0.00) | (1.45 to 2.72) | 1.99 (0.00) | (1.45 to 2.73) | 1.99 (0.00) | (1.45 to 2.73) | 1.99 (0.00) | (1.46 to 2.73) |
| Child sex (Ref: Male) | | | | | | | | | | |
| Female | | | 0.88 (0.00) | (0.80 to 0.95) | 0.88 (0.00) | (0.80 to 0.96) | 0.88 (0.00) | (0.80 to 0.96) | 0.88 (0.00) | (0.80 to 0.96) |
| Child sickness type (Ref: diarrhoea) | | | | | | | | | | |
| Fever | | | 1.17 (0.02) | (1.02 to 1.33) | 1.17 (0.02) | (1.03 to 1.34) | 1.17 (0.02) | (1.03 to 1.34) | 1.17 (0.02) | (1.03 to 1.34) |
| Acute respiratory infection | | | 1.77 (0.00) | (1.53 to 2.06) | 1.78 (0.00) | (1.54 to 2.06) | 1.78 (0.00) | (1.53 to 2.06) | 1.79 (0.00) | (1.53 to 2.06) |
| Mother education (Ref: no education) | | | | | | | | | | |
| Primary | | | | | 1.29 (0.00) | (1.12 to 1.49) | 1.29 (0.00) | (1.13 to 1.49) | 1.30 (0.00) | (1.13 to 1.49) |
| Secondary or higher | | | | | 1.65 (0.00) | (1.38 to 1.97) | 1.65 (0.00) | (1.38 to 1.97) | 1.65 (0.00) | (1.38 to 1.97) |
| Household wealth quintile (Ref: Q1 poorest) | | | | | | | | | | |
| Q2 | | | | | 0.91 (0.24) | (0.07) | 0.92 (0.38) | (0.77 to 1.10) | 0.92 (0.38) | (0.77 to 1.10) |
| Q3 | | | | | 0.96 (0.60) | (0.08) | 0.98 (0.82) | (0.82 to 1.17) | 0.98 (0.82) | (0.82 to 1.17) |
| Q4 | | | | | 0.99 (0.87) | (0.08) | 1.01 (0.91) | (0.84 to 1.21) | 1.01 (0.91) | (0.84 to 1.21) |
| Q5 | | | | | 1.00 (0.96) | (0.08) | 1.02 (0.84) | (0.85 to 1.22) | 1.01 (0.83) | (0.85 to 1.22) |
| Household rural residence | | | | | | | 0.95 (0.57) | (0.80 to 1.13) | 0.95 (0.56) | (0.80 to 1.13) |
| Observations | 9701 | | 9701 | | 9695 | | 9695 | | 9695 | |

*95% CI: 95% Confidence Intervals

†In the main model (n=568), the analysis was restricted to facilities offering sick child healthcare that matched (within 50km) to a household sampled in the Multiple Indicator Cluster Survey with a recently ill child.

Future research is needed to validate and extend these findings in other country settings. Malawi has been a leader in SSA, demonstrating strong political will to implement evidence-based interventions that can improve MCH. The availability of the geocoded health facilities and population data enabled us to match the health system facility survey data and the population data in. Other countries could take similar measures to permit matching of health system and household data to obtain better insights in how health systems influence health and care-seeking. These measures would aid in policy determinations, to evaluate whether services supplied by the health system can actually meet with the demands of the population, and quality of care can satisfy the population's needs and promote patient care-seeking behaviours that reduce preventable deaths. In addition, the facility quality index needs to be validated in different country settings, encompassing higher mortality burdens and different health system capacities, to strengthen the generalisability of the results.

As an important component of human capital, health can contribute positively to a nation's economic development.[54] At the same time, achieving an excellent state of health is an intrinsic part of the goals of social development, as well as an essential factor in an individual's well-being. Good quality healthcare is thus both an intrinsic good that can promote health outcomes and a driver of utilisation; to achieve these ends, both structures and processes of care need to improve.

**Acknowledgements** Earlier versions of this paper were presented during several seminars at the Harvard Global Health Institute, the Harvard T.H. Chan School of Public Health, and the Fifth Global Symposium on Health Systems Research, where valuable comments were received from seminar participants, including Stéphane Verguet, Michael Reich and Sara Singer. The authors thank Dennis Lee and Fei Carnes for assistance with geographical linking. The authors also acknowledge Humphreys Nsona for his input on the initial study idea and assistance with the geospatial data.

**Contributors** MEK conceptualised the study. HHL and LL curated the data. LL conducted the formal analysis. HHL contributed to study design. LL visualised the data results and wrote the original draft. LL, HHL, MJ and MEK reviewed and contributed to the editing of the manuscript. All authors approved the final manuscript submitted.

**Funding** This work is partially supported by the Bill & Melinda Gates Foundation (OPP1161450, Kruk). The funding sources had no role in this work.

**Map disclaimer** The depiction of boundaries on the map(s) in this article do not imply the expression of any opinion whatsoever on the part of BMJ (or any member of its group) concerning the legal status of any country, territory, jurisdiction or area or of its authorities. The map(s) are provided without any warranty of any kind, either express or implied.

**Competing interests** None declared.

**Patient consent for publication** Not required.

**Ethics approval** The Harvard University Human Research Protection Program deemed this analysis of secondary data exempt from human subject review.

**Provenance and peer review** Not commissioned; externally peer reviewed.

**Data sharing statement** Data are available in a public, open access repository.

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
