## [Reviewer comments · BMJ Open]

ARTICLE DETAILS

TITLE (PROVISIONAL)	Exploring the association between sick child healthcare utilization and health facility quality in Malawi: A Cross-Sectional Study
AUTHORS	LIU, LINGRUI; Leslie, Hannah; Joshua, Martias; Kruk, Margaret

VERSION 1 – REVIEW

REVIEWER	Sonia Lewycka Oxford University Clinical Research Unit, Ha Noi, Viet Nam
REVIEW RETURNED	12-Mar-2019

GENERAL COMMENTS	This paper makes a novel linkage between household-level and facility-level datasets to explore quality of healthcare as a determinant of healthcare utilisation. The methods, results and discussion are all appropriate, but I think the authors need to acknowledge that survey data on symptoms of illness does not have the specificity to identify all children who require formal healthcare. However, this does not detract from the fact that the authors have identified an association between quality of healthcare and healthcare utilisation. The non-specificity of the definition of the included sub-population (i.e. those who actually need formal healthcare), plus not being able to directly link individuals to the actual health facility they used (i.e. the outcome measure - healthcare utilisation) just adds noise to the model. The authors should address this in the study limitations. Another area for clarification is whether pharmacies should be included. MICS includes pharmacies for source of ORS and zinc, but not for healthcare providers for diarrhoea care (and ARI/fever?). So probably pharmacies should not be included when considering formal healthcare utilisation. How about community health workers and mobile/outreach clinics? These are included in MICS, but not covered by SPA. Again, people seeking care from these sources should probably not be included in a model assessing the influence of health facility quality on healthcare utilisation, since accessing care from a community health worker or outreach has nothing to do with the quality of the nearest facility. Creating a quality of care index simply by adding the presence of each component gives each component the same weight. For future analysis, it would be interesting to do some further exploration of which components are more strongly related to healthcare utilisation. This could be done using PCA or latent class analysis Another area for further analysis - It would be interesting to do models separately for ARI, fever and diarrhoea. Care-seeking from
--

	informal sources for diarrhoea (ORS) and to a lesser extent fever (over the counter LA) may be a factor in decisions about when to seek formal healthcare. Looking at healthcare utilisation particularly for those with ARI alone would be useful. Minor points for clarity  - Title and elsewhere in the document - Rather than using the term "sick child care utilization" it would be clearer to specify "sick child healthcare utilization". Care may not be healthcare related. - I was getting a bit confused about the difference between MES and MICS, but I think they are the same. It would be good to choose one acronym and stick with it throughout the paper. - Please provide 95% confidence intervals and actual p-values in Table 3 rather than using SD and * The reviewer provided a marked copy with additional comments. Please contact the publisher for full details.
--	---

REVIEWER	Diwakar Mohan Johns Hopkins University
REVIEW RETURNED	23-Mar-2019

GENERAL COMMENTS	Thank you for the opportunity to review your study. The study links two nationally representative surveys – SPA and MDGE – to understand the association between quality of sick child and utilization of care. General comment – The manuscript is very well written with no need for English language editing. Some grammar / spelling check needed (highlighted in comment boxes in accompanying PDF) Overall, a well thought out and appropriately designed study. I have raised a few points below to invite responses from the authors with a complete understanding that this may not change the methodology of the study or the findings presented. Major issues  1. When used in DHS / MICS like surveys, EA geocodes are displaced by 5km or so for reasons of confidentiality. I assume this is the case for this survey as well. If the geocodes are displaced by such distances, using the centroid for linkage may not be correct. Can the authors confirm that they have used undisplaced geo-coordinates for the EAs sampled? This would mean that the NSO was able to provide the EA numbers corresponding to the 2008 census for each MICS sampled cluster. Can the authors also confirm this is in line with the ethical requirements of the Household survey used? 2. There is no presentation of the check for the validity of the linking method. An example of a check could be cross-tabulating the type of facility used for the care seeking episode against the type of facility linked from the SPA survey. This will provide an estimate of how well the linkage works. 3. Staffing & Management domain appears to be over-weighted by the supervisory variable – 5 of the 9 items are supervisory items and crowd out other important indicators.
---

	Training in handling sick children, in my opinion, is the most important indicator of readiness. 4. Multiple imputation assumes the data is missing at random. Is it the contention of authors that the non-observation of sick child visits at certain facilities was due to randomness? Also, if imputation is justified, then the method of imputation and the variables used for the imputation should be described. I would posit that the reason for non-observation of sick child visits was the low utilization at these facilities which is an outcome under consideration. To impute entire lists of process indicator values for multiple facilities might be biased. 5. It is not clear from the description provided whether multilevel models have been used. Considering the hierarchical nature of the data – observations nested within EAs linked to facilities with possibility of multiple EAs linked to a single facility – it might be prudent to consider multilevel models. 6. Considering that facility characteristics like type (hospital / health center) and management (public / CHAM/NGO) are important considerations for the quality, they should be included as part of the models as controlling factors. Minor revisions Line 197 – It would be good to know more about the rescaling process. At the moment, it is not very clear on how the rescaling was done. Maybe the sentence needs to be split between averaging structural indicators and averaging process indicators. Line 276 – are the 352 dropped because of the 50km distance condition? This means that close to 40% of the facilities have been dropped from the analysis. I feel this is too large a number to drop from analysis without biasing the results. One possibility for this lack of linkage is the sampling process. To address Can you compare the characteristics of the facilities included and exclude stratified by urban and rural? Line 308 – the overall quality score is different from the number presented in table 1. Figure 1 – What is the geographic area for which the density is being presented in the figure? Table 2 – The third column is redundant since it is the difference between the first and second. More importantly, it would be good to know if any characteristics are differentially distributed between the sick children and those who sought care. Table 3 – Please verify the coefficients for the age 2month-1 year category. Looking at the heterogeneity in the different types of illnesses, it might be interesting to run the full models (4 & 5) separately for each type of illness. The reviewer provided a marked copy with additional comments. Please contact the publisher for full details.
--	---

VERSION 1 – AUTHOR RESPONSE

Reviewer: 1

Reviewer Name: Sonia Lewycka

Institution and Country: Oxford University Clinical Research Unit, Ha Noi, Viet Nam Please state any competing interests or state 'None declared': None declared

Please leave your comments for the authors below

This paper makes a novel linkage between household-level and facility-level datasets to explore quality of healthcare as a determinant of healthcare utilisation.

Response: We thank reviewer 1 for the favorable comments and support. We have made every effort to address the various comments and suggestions of the reviewer. The line numbers listed in the response sheet are the line numbers in the version with marked changes.

The methods, results and discussion are all appropriate, but I think the authors need to acknowledge that survey data on symptoms of illness does not have the specificity to identify all children who require formal healthcare. However, this does not detract from the fact that the authors have identified an association between quality of healthcare and healthcare utilisation. The non-specificity of the definition of the included sub-population (i.e. those who actually need formal healthcare), plus not being able to directly link individuals to the actual health facility they used (i.e. the outcome measure - healthcare utilisation) just adds noise to the model. The authors should address this in the study limitations.

Response: We thank the reviewer for this comment. We included an acknowledgement of this limitation in the "Strengths and limitations of this study" section and in the "limitation" section. Line: 67-69. Line: 466-470.

Another area for clarification is whether pharmacies should be included. MICS includes pharmacies for source of ORS and zinc, but not for healthcare providers for diarrhoea care (and ARI/fever?). So probably pharmacies should not be included when considering formal healthcare utilisation. How about community health workers and mobile/outreach clinics? These are included in MICS, but not covered by SPA. Again, people seeking care from these sources should probably not be included in a model assessing the influence of health facility quality on healthcare utilisation, since accessing care from a community health worker or outreach has nothing to do with the quality of the nearest facility.

Response: We thank the reviewer for this consideration. In this study, to define "health facilities", we considered the definition from MICS survey regarding health facility types from where caregivers brought sick children to seek care, and also the facility types (including hospitals, health centers, clinics, health posts, dispensaries) covered by SPA. In addition, for the reason as the reviewer mentioned here that the diarrhea care is not available at pharmacies, we didn't include pharmacies. For "outreach clinics", based on our consultation with our coauthor in Malawi, this type of health facility can be considered as "dispensaries" which is covered by SPA. Therefore, we included "outreach clinics" as one type of health facilities at community-level, which accounts for about 4% of our analytic sample. In this revised version, we added a clarification of the health facilities in this study in line 222-225 and corrected the wording to use SPA term "dispensaries" to avoid confusion.

Creating a quality of care index simply by adding the presence of each component gives each component the same weight. For future analysis, it would be interesting to do some further exploration of which components are more strongly related to healthcare utilisation. This could be done using PCA or latent class analysis

Response: We thank the reviewer for this idea for a future study to incorporate weights in the quality of care index. In this study, we relied on WHO SARA and IMCI guidelines to construct the quality

measurement, which is intended to capture all needed elements of a health visits, rather than expressing a latent construct. That said, a future study could investigate the different weights of the components in the quality of care which are more strongly related to healthcare utilization, particularly in LMIC settings.

Another area for further analysis - It would be interesting to do models separately for ARI, fever and diarrhoea. Care-seeking from informal sources for diarrhoea (ORS) and to a lesser extent fever (over the counter LA) may be a factor in decisions about when to seek formal healthcare. Looking at healthcare utilisation particularly for those with ARI alone would be useful.

Response: We thank the reviewer for sharing this future analysis idea. In this paper, we mainly focused on the research question that whether the utilization of care for sick children is influenced by the overall quality of sick child healthcare. We would consider the reviewer's suggested future analysis idea in a next paper focusing on the specific illness conditions which can be built on the finding of our paper about the association between utilization and quality in sick child healthcare in Malawi.

Minor points for clarity

- Title and elsewhere in the document - Rather than using the term "sick child care utilization" it would be clearer to specify "sick child healthcare utilization". Care may not be healthcare related.

Response: We thank the reviewer for this suggestion and we revised the wording of this term throughout the manuscript in this revision.

- I was getting a bit confused about the difference between MES and MICS, but I think they are the same. It would be good to choose one acronym and stick with it throughout the paper.

Response: We thank the reviewer for this suggestion and we revised the draft to use MICS consistently throughout the paper to avoid confusion.

- Please provide 95% confidence intervals and actual p-values in Table 3 rather than using SD and *

Response: We thank the reviewer for this suggestion and we revised Table 3 accordingly.

Please find more specific comments in the attached PDF (Liu_BMJOpen_child care utilisation quality Malawi_SL)

Response: we revised our manuscript at the places where the reviewer commented. Below are a few more prominent comments for which we list out our responses.

Line 88: How do we know it is not sufficient? Perhaps people are buying over the counter treatments such as ORS and LA. In some SSA countries malaria RDTs can be done outside of the formal healthcare system. I'm not sure whether or not this is the case now in Malawi. I don't think there is information in MICS on how many children with fever used an RDT to test for malaria anyway.

Response: We thank the reviewer for this consideration. We acknowledged that MICS report did not provide the actual number of children with fever used RDT to test and confirm malaria. The proportions of sick children under five with symptoms of ARI, fever, and diarrhea who were taken to a health provider for treatment within 48 hours of symptom onset reported by MICS can indicate an overall pattern of sick children (their caregivers) utilizing existing health system.

Line 261 Were those not offering sick child care excluded? Therefore state the final number of facilities included

Response: We excluded the facilities which did not provide sick child healthcare services (line 270-271). The final number of facilities included in our analytic sample is 568, which we reported in the original version (located in line 294-295 in this revised version).

Table 1.

Need to show or at least state the other categories here - i.e. are they urban and private?

Response: Thank you for this comment. We include rural and urban facilities. The authority of health facilities is either public or private. We revised the table rows.

Table 2

Would be more interesting to have row percentages here - e.g. % of mothers with no education who sought care vs sought no care. Except for the first column, where column percentages are more useful.

Or, you could put the row n and % as additional columns in Table 3.

Response: Thank you for this comment. In this revision, we reported the row % for the group who sought for care during illness and the group who did not seek for care during illness.

Line 342 the odds of utilizing formal health facilities increases by 56% (AOR: 1.56, SE: 0.31, 95% CI: 1.04, 2.28) for every hundred percent increase in the overall quality. This doesn't make sense. Can quality be over 100%?

Response: Thank you for this comment. The quality score is a proportion with a theoretical minimum of 0 and maximum of 1. Therefore, the OR represents the change from absolute minimum of 0 quality to maximum of 1 (perfect) quality. We revised the wording in the manuscript.

Line 390.

Not clear - is this four out of every five children with pneumonia diagnosed as not having pneumonia, or those with pneumonia misdiagnosed as having pneumonia when they didn't have pneumonia, or just that guideline adherence was only for 4 out of 5 children? Is this a problem of the health-workers, or the guidelines? In the absence of diagnostic tests, it would be difficult to identify all pneumonia cases with a high degree of accuracy.

Response: Thank you for this comment. We cited this finding from the reference, Uwemedimo OT, Lewis TP, Essien EA, et al. Distribution and determinants of pneumonia diagnosis using Integrated Management of Childhood Illness guidelines: a nationally representative study in Malawi. *BMJ Glob Health* 2018;3:e000506. doi:10.1136/bmjgh-2017-000506. Using the data from the 2013 Malawi SPA, it found that of the 3136 children in its sample at least 2 months old, 573 (18%) met the IMCI-based criteria for pneumonia diagnosis based on re-examination. It also found that, only about 118 (21%) of 573 children who met the IMCI criteria for pneumonia in Malawi were correctly diagnosed. Of the 455 children meeting the case definition of pneumonia who were not diagnosed with it (ie, had a "missed diagnosis"), 212 (46%) were diagnosed with an upper respiratory infection and 158 (34%) were diagnosed with malaria; older children were more likely to be diagnosed with malaria than children under one (36.4% vs 21.4%, $P < 0.001$, data not shown). It found that better sensitivity, interventions like training and supervision are associated with the guideline adherence. "Better sensitivity is associated with provider type and higher adherence to IMCI. Existing

interventions such as training and supportive supervision are associated with higher guideline adherence, but are insufficient to meaningfully improve sensitivity.”

Line 419-420

Be consistent when referring to MICS or MES. This is not the health system data, but the healthcare utilisation data?
clinically relevant?

Response: We thank the review for this comment. We revised the language to use MICS consistently throughout the manuscript.

Line 423

There is no mortality data linked to healthcare utilisation data in the MICS datasets, but you could link the MICS U5MR by cluster with the health facility data in the same way.

Response: We thank the review for this comment. We have conducted this analysis in another separate study currently underway.

Reviewer: 2

Reviewer Name: Diwakar Mohan

Institution and Country: Johns Hopkins University

Please state any competing interests or state 'None declared': None declared

Please leave your comments for the authors below

Please find my comments in the attached word document (bmj_malawi_qoc) and a few small edits in the PDF (bmj_malawi_qoc).

Reviewer comments

Exploring the association between sick child care utilization and health facility quality in Malawi: A Cross-Sectional Study

Thank you for the opportunity to review your study.

The study links two nationally representative surveys – SPA and MDGE – to understand the association between quality of sick child and utilization of care.

General comment – The manuscript is very well written with no need for English language editing. Some grammar / spelling check needed (highlighted in comment boxes in accompanying PDF)

Overall, a well thought out and appropriately designed study. I have raised a few points below to invite responses from the authors with a complete understanding that this may not change the methodology of the study or the findings presented.

Response: We thank reviewer 2 for your constructive comments and support to improve our paper manuscript. We have made every effort to address the various comments and suggestions of the reviewer. The line numbers listed in the response sheet are the line numbers in the version with marked changes.

Major issues

1. When used in DHS / MICS like surveys, EA geocodes are displaced by 5km or

so for reasons of confidentiality. I assume this is the case for this survey as well. If the geocodes are displaced by such distances, using the centroid for linkage may not be correct.

Can the authors confirm that they have used undisplaced geo-coordinates for the EAs sampled? This would mean that the NSO was able to provide the EA numbers corresponding to the 2008 census for each MICS sampled cluster.

Can the authors also confirm this is in line with the ethical requirements of the Household survey used?

Response: We thank the reviewer for this thoughtful comment. The NSO was able to provide the EA centroid (not household location) without displacement. This analysis was reviewed by Harvard IRB and provided exempt approval due to non-identifiability of individuals in the study.

2. There is no presentation of the check for the validity of the linking method. An example of a check could be cross-tabulating the the type of facility used for the care seeking episode against the type of facility linked from the SPA survey. This will provide an estimate of how well the linkage works.

Response: We thank the reviewer for this comment. In our design of the linking approaches, we considered to test the validity of our linking method and therefore included a series of sensitivity analyses to understand the robustness of our results. In this revision, we adopted the reviewer’s suggestion and ran a cross-tabulation of the type of the facility used for the care seeking against the type of facility linked from the SPA survey in our analytic sample (main matching strategy). See the results below. There are some missing values for the facility type reported in the MICS dataset. However, the distribution of the facility type from MICS (care-seeking) and SPA (health facility) were similar. For hospital utilization and to some extent for health centers, we would expect to see a substantial degree of bypassing, i.e., people who report using hospitals are likely to have a lower level clinic closer to them.

mics_factype	SPA factype					Total
	Hospital	Health center	clinic	Health post	Other	
Government hospital	340	229	355	1	33	958
Government health cen	281	2,078	428	16	132	2,935
Government health pos	15	113	23	2	12	165
Village health worker	91	460	83	3	17	654
Mobile / outreach cli	12	36	10	0	2	60
Other public	0	6	1	0	1	8
Private hospital / cl	30	100	86	0	9	225
Private physician	0	6	3	0	0	9
Private pharmacy	61	177	52	0	10	300
.	662	2,569	957	28	171	4,387
Total	1,492	5,774	1,998	50	387	9,701

3. Staffing & Management domain appears to be over-weighted by the supervisory variable – 5 of the 9 items are supervisory items and crowd out other important indicators. Training in handling sick children, in my opinion, is the most important indicator of readiness.

Response: We thank the reviewer for this comment. We acknowledged the data limitations and extracted the index items available from the Malawi SPA dataset with reliance on the WHO SARA and IMCI. There is one item under “Staffing & Management”- “staff training on child health services” which is what the reviewer suggested.

4. Multiple imputation assumes the data is missing at random. Is it the contention of authors that the non-observation of sick child visits at certain facilities was due to randomness? Also, if imputation is justified, then the method of imputation and the variables used for the imputation should be described. I would posit that the reason for non-observation of sick child visits was the low utilization at these facilities which is an outcome under consideration. To impute entire lists of process indicator values for multiple facilities might be biased.

Response: We thank the reviewer for this comment.

Because the assumption required for multiple imputation is that missingness is random conditional on the covariates, we included covariates in our imputation: facility type (e.g., central hospital, district hospital, health centers, etc.), facility managing authority (e.g., government, private, NGO, mission/faith-based, etc.), districts, and urban/rural. Our assumption is that missingness within a stratum of these covariates, for instance public dispensaries in a rural area of a given district, is random. We imputed the quality item scores for the facilities which provide sick child healthcare but did not have observations and merged the kids to their nearest facility with all the 920 facilities which provide sick child healthcare. It resulted in an analytic sample of 9701 sick children with 568 matched facilities which provide sick child healthcare. In the imputation, we only impute the variables which had missing values and also exclude the variables which had missing values due to skip patterns, for instance whether diagnostic supplies were present if a facility did not have laboratory services at all (which we set to 0 for the purpose of this analysis). For instance, medical storage/medical dates/stock update is not asked if there is no pharmacy. We included these details in our statement of the method of imputation in the method section (line 211- 215) in this revision and also acknowledged this in the limitation section (line 470-475).

5. It is not clear from the description provided whether multilevel models have been used. Considering the hierarchical nature of the data – observations nested within EAs linked to facilities with possibility of multiple EAs linked to a single facility – it might be prudent to consider multilevel models.

Response: We thank the reviewer for this comment. While it would have been possible to use a multilevel model, we used a logistic model as we were not primarily interested in apportioning variance. However, we revised our models and used clustered standard errors (cluster at the EA level) to account for the non-independence of observations within EAs. We included this detail in line 259-260.

6. Considering that facility characteristics like type (hospital / health center) and management (public / CHAM/NGO) are important considerations for the quality, they should be included as part of the models as controlling factors.

Response: We thank the reviewer for this comment. The research question in this paper is whether the quality of the sick child healthcare available to parents affect their seeking care when they recognize their children's illness, regardless of the type of facility. Given the nature of our research question, we did not include the type and management of facility as controlling factors. Considering the reviewer's comment, we ran additional model that included type (hospital or not hospital) and management (public or private) of facility. This resulted in minor changes in the magnitude and the significance of the results. However, the main conclusions that better quality of sick child healthcare is associated with higher odds of utilization of sick child healthcare services has remained unchanged. Including the facility type as an additional controlling factor, the association with overall quality was stronger (AOR=1.70, 95% CI: 1.04, 2.77), while including both facility type and management of facility, the association with overall quality was weaker (AOR=1.24, 95% CI: 0.75, 2.06).

Minor revisions

Line 197 – It would be good to know more about the rescaling process. At the moment, it is not very clear on how the rescaling was done. Maybe the sentence needs to be split between averaging structural indicators and averaging process indicators.

Response: We thank the reviewer for this comment. We split the long sentence for a clear narrative in this revision. Line 201-209, as below.

We first averaged structural indicators within facility. We then averaged process indicators within observation. Further, we averaged these clinical observations for sick child visits within each facility to obtain the facility level process quality score, using a rescaled weight for each clinical observation to reflect the sampling probability of patients within facilities. We further averaged the facility-level structure and process quality score to arrive at a facility-level overall quality score to measure average performance on all indicators.

Line 276 – are the 352 dropped because of the 50km distance condition? This means that close to 40% of the facilities have been dropped from the analysis. I feel this is too large a number to drop from analysis without biasing the results.

One possibility for this lack of linkage is the sampling process. To address Can you compare the characteristics of the facilities in included and exclude stratified by urban and rural?

Response: We thank the reviewer for this comment. The reason the 352 health facilities were dropped was not because of the 50 km distance condition. They were dropped because they were not matched as the nearest facility to any child's household based on road traveling distance. Among the 920 health facilities, by our main matching strategy, only the nearest facility to the child's household was matched with the sick child.

Regarding the representativeness, the MICS is a representative sample, and we included the facilities that served individuals sampled in the MICS. Although this is not guaranteed to be a certain proportion of all facilities and may not be representative in terms of facility number, the health facilities in our analytic sample should be representative in terms of facilities serving the population. Small, rural facilities that serve just a few clusters of people are only included if that particular rural cluster was selected for the MICS.

Line 308 – the overall quality score is different from the number presented in table 1.

Response: We apologize for this typo. We corrected the numbers in Table 1. We obtained the means of the imputed structural quality score, imputed process quality, and the imputed overall quality score, from Stata calculation (mi estimate: mean xxx).

Figure 1 – What is the geographic area for which the density is being presented in the figure?

Response: Figure 1 presents the location of all Malawi SPA health facilities and the location of SPA health facilities which are included in our analytic sample, as well as the population density in Malawi.

Table 2 – The third column is redundant since it is the difference between the first and second. More importantly, it would be good to know if any characteristics are differentially distributed between the sick children and those who sought care.

Response: We thank the reviewer for this comment. We revised our Table 2. It now reports the row % of the distribution of the characteristics between the sick children who sought care and those who did not seek care when had the illness.

Table 3 – Please verify the coefficients for the age 2month-1 year category.

Response: We thank the reviewer for this comment. We revised and updated our Table 3 in this revision.

Looking at the heterogeneity in the different types of illnesses, it might be interesting to run the full models (4 & 5) separately for each type of illness.

Response: We thank the reviewer for sharing this future analysis idea. In this paper, we mainly focused on the research question that whether the utilization of sick child healthcare is influenced by the overall quality of sick child healthcare. The separate assessment by illness type was not a goal of this paper. We would consider the reviewer's suggested future analysis idea in a next paper focusing on the specific illness conditions which can be built on the finding of our paper about the association between utilization and quality in sick child healthcare in Malawi.

Considering the reviewer's comments, we run full models separately for each type of illness and would like to present results in this response sheet. This resulted in changes in the magnitude and the significance of the results (the AOR, 95% CI for quality variables). The AOR from the final models (in analyses separated by child illness type), the association with overall quality was stronger for Fever illness (AOR=1.80, 95% CI: 1.00, 3.25) and weaker for diarrhea (AOR= 1.41, 95% CI: 0.40, 4.89) and ARI (AOR= 1.46, 95% CI: 0.58- 3.67). However, they all suggest a positive association between quality and utilization of sick child healthcare services.

VERSION 2 – REVIEW

REVIEWER	Sonia Lewycka University of Auckland/University of Oxford
REVIEW RETURNED	16-May-2019

GENERAL COMMENTS	Thanks for addressing the points I raised. This is an interesting analysis and could be extended in several ways as I previously suggested. The only minor point I would address is that the text in lines 375-6 is still a bit unclear. Why not just say: "the odds of utilizing formal health facilities increases with increasing quality (AOR: 1.66, 95% CI: 1.04, 2.63)"?
--

REVIEWER	Diwakar Mohan Johns Hopkins University
REVIEW RETURNED	29-May-2019

GENERAL COMMENTS	Thank you for taking the time to address the questions raised in my previous review. The manuscript addresses the relationship between quality of care and utilization of care seeking for sick children in Malawi. Minor suggestions  Line 391-392 – Should this be severity of illness or type of illness? I assume the inference is being made about care seeking for fever and pneumonia vs diarrhea.
--

2. Figure 1 – It is not clear from the figure what the polygons represent. I doubt if these are the boundaries of enumeration areas since the size of the EAs is quite small. Also, the Malawi lake seems to be incorporated into the polygons. The polygons look more like sub-district administrative unit rather than an EA. Please confirm and, if necessary, change the text in the figure and within the body of the manuscript before publication.

I have attached an image showing the EA boundaries in Malawi. Your polygons are far too big to be EAs.

VERSION 2 – AUTHOR RESPONSE

Reviewer: 1

Reviewer Name: Sonia Lewycka

Institution and Country: Oxford University Clinical Research Unit, Ha Noi, Viet Nam Please state any competing interests or state 'None declared': None declared

Please leave your comments for the authors below

Thanks for addressing the points I raised. This is an interesting analysis and could be extended in several ways as I previously suggested.

Response: We thank reviewer 1 for the favorable comments and support. We have addressed the comments and suggestions of the reviewer.

Minor revisions

The only minor point I would address is that the text in lines 375-6 is still a bit unclear. Why not just say: "the odds of utilizing formal health facilities increases with increasing quality (AOR: 1.66, 95% CI: 1.04, 2.63)"?

Response: We thank reviewer 1 for this suggestion. We revised the line 368- 370.

In the fully specified model (Model 4), the odds of utilizing formal health facilities increases with increasing quality (AOR: 1.66, 95% CI: 1.04, 2.63).

Reviewer: 2

Reviewer Name: Diwakar Mohan

Institution and Country: Johns Hopkins University

Please state any competing interests or state 'None declared': None declared

Please leave your comments for the authors below

Please find my comments in the attached word document (bmj_malawi_qoc) and a few small edits in the PDF (bmj_malawi_qoc).

Reviewer comments

Exploring the association between sick child care utilization and health facility quality in

Malawi: A Cross-Sectional Study

Thank you for the opportunity to review your study.

The study links two nationally representative surveys – SPA and MDGE – to understand the association between quality of sick child and utilization of care.

General comment – Thank you for taking the time to address the questions raised in my previous review. The manuscript addresses the relationship between quality of care and utilization of care seeking for sick children in Malawi.

Response: We thank reviewer 2 for your constructive comments and support to improve our paper manuscript. We have made every effort to address the various comments and suggestions of the reviewer.

Minor revisions

1. Line 391-392 – Should this be severity of illness or type of illness? I assume the inference is being made about care seeking for fever and pneumonia vs diarrhea.

Response: We thank the reviewer for this comment. We revised the text in line 384.

This indicates that the type of childhood illness (i.e., diarrhea, fever, or ARI) is associated with motivating caregivers to utilize health facilities.

2. Figure 1 – It is not clear from the figure what the polygons represent. I doubt if these are the boundaries of enumeration areas since the size of the EAs is quite small. Also, the Malawi lake seems to be incorporated into the polygons. The polygons look more like sub-district administrative unit rather than an EA. Please confirm and, if necessary, change the text in the figure and within the body of the manuscript before publication.

Response: We thank the reviewer for this comment. We revised the map and the text in line 295- 296.

We revised the map using the following data sources:

- Population density: Worldpop.org open access
<https://www.worldpop.org/geodata/summary?id=123>

CITATION: WorldPop. 2017. Malawi 100m Population, Version 2. University of Southampton. DOI: 10.5258/SOTON/WP00538.

- MICS enumeration areas: Malawi National Statistical Office.
- Malawi shapefile and health facility location: The DHS Program

In this revised version, we revised our map using the latest population density estimates data from the Worldpop.org for numbers of people per hectare for 2015. The revised map shows the 2013 MICS enumeration areas and the district boundaries. In Malawi, the full census included about 12000 enumeration areas and about 1100 of these were selected for the MICS. We also included a statement of the data source on the map in Figure 1 and noted in the methods the software program used to create the map.

Figure 1 shows the geographic distribution of all 2013 SPA health facilities and the ones included in our analytic sample, the 2013 MICS enumeration areas, as well as the population density in Malawi.